# Induction of Labor in Women with Previous Cesarean Section and Unfavorable Cervix: A Retrospective Cohort Study

**DOI:** 10.3390/healthcare11040543

**Published:** 2023-02-12

**Authors:** Chiara Germano, Ilenia Mappa, Antonella Cromi, Enrico Busato, Maddalena Incerti, Andrea Lojacono, Giuseppe Rizzo, Rossella Attini, Lodovico Patrizi, Alberto Revelli, Bianca Masturzo

**Affiliations:** 1Department of Obstetrics and Gynaecology 2U, Sant’Anna Hospital, University of Turin, 10126 Turin, Italy; 2Obstetrics and Gynaecology Department, Infermi Hospital, University of Turin, 10124 Turin, Italy; 3Division of Maternal Fetal Medicine Ospedale Cristo Re, University of Roma Tor Vergata, 00133 Rome, Italy; 4Obstetrics and Gynaecology Department, Filippo del Ponte Hospital, University of Insubria, 21100 Varese, Italy; 5Obstetrics and Gynaecology Department, Ca’ Foncello Hospital, 31100 Treviso, Italy; 6Obstetrics and Gynaecology Department, San Gerardo Hospital, 20900 Monza, Italy; 7Obstetrics and Gynaecology Department, Spedali Civili Hospital, University of Brescia, 25123 Brescia, Italy; 8Department of Obstetrics and Gynecology Fondazione Policlinico Tor Vergata, University of Roma Tor Vergata, 00133 Rome, Italy

**Keywords:** cesarean section, trial of labor after cesarean section, cervical ripening balloon, unfavorable cervix

## Abstract

*Background:* The efficacy and safety of a cervical ripening balloon (CRB) in women with a previous cesarean section (CS) and unfavorable Bishop score are still controversial. *Methods:* A retrospective cohort study was performed across six tertiary hospitals from 2015 to 2019. Women with one previous transverse CS, singleton cephalic term pregnancy and BS < 6 were included if submitted to labor induction with a CRB. The main outcome was the rate of vaginal birth after cesarean (VBAC) after CRB ripening. Secondary outcomes were abnormal composite fetal and maternal outcomes. *Results:* Of the 265 women included, 57.3% had successful vaginal birth. Augmentation improved vaginal delivery (32.2% vs. 21.2%). Intrapartum analgesia was associated with an increased VBAC rate (58.6% vs. 34.5%). Maternal BMI ≥30 and age ≥40 years increased emergency CS rate (11.8% vs. 28.3% and 7.2 vs. 15.9%). Composite adverse maternal outcome occurred in 4.8% of CRB group women and increased to 17.6% when associated with oxytocin. Uterine rupture occurred in one case (0.4%) in the CRB–oxytocin group. Poorer fetal outcome occurred after emergency CS, if compared to successful VBAC (12.4% vs. 3.3%). *Conclusions:* In women with a previous CS and unfavorable Bishop score, induction of labor with a CRB can be considered safe and effective.

## 1. Introduction

The best delivery mode for women with a previous cesarean section (CS) is still a matter of discussion, and clinical practice varies worldwide. Elective repeated cesarean section (ERCS) represents approximately 28% of all CSs in the UK [1] and 30–50% in the United States [2], where the indication “previous cesarean section” relevantly contributes to increased CS rate. ERCS is frequently chosen because a trial of labor after cesarean section (TOLAC) may fail or have serious complications, e.g., uterine rupture [3]; on the other side, however, women who experience vaginal birth after previous cesarean section (VBAC) are less likely to face birth-related morbidity such as blood transfusion, uterine rupture, unplanned hysterectomy, and admission to an intensive care unit than women who have ERCS [4].

The objective of VBAC can be obtained without disregarding safety by accurately selecting candidates undergoing TOLAC [5], and by choosing a well-customized method of induction. In 2015, the Royal College stated that TOLAC might be offered to women with one previous CS with lower segment incision, singleton term pregnancy, cephalic presentation, with or without history of previous vaginal delivery [6]. The accurate selection of patients is mandatory to reduce the risk of TOLAC failure [7,8], and contributes to increasing the success rate and decreasing negative maternal–fetal outcomes.

Recently, international guidelines have clearly endorsed labor induction with mechanical devices, especially in the case of an unfavorable cervix, due to the lower risk of uterine rupture compared to pharmacological induction with oxytocin and/or prostaglandins [6,9,10,11]. The guidelines underlined, however, that there are still too few studies dealing with the use of a cervical ripening balloon (CRB) for labor induction after CS, particularly when the Bishop score is unfavorable. Therefore, the objective of the present study was to retrospectively evaluate the efficacy and safety of a CRB for labor induction in a fair-sized cohort of women with a previous CS and unfavorable cervix.

## 2. Materials and Methods

This retrospective, multicenter cohort study was performed in six Italian tertiary hospitals in a 5-year period (2015–2019), including women with singleton term pregnancy, cephalic presentation, and previous CS (Robson Classification—Group 5). Exclusion criteria were: spontaneous labor onset, more than one uterine scar, Bishop score > 6, declined labor induction, any other contraindication for vaginal delivery (e.g., placenta previa, non-cephalic presentation, etc.), fetal anomalies, and stillbirth (Figure 1).

Data were collected retrospectively and anonymously in a shared database; each enrolled patient was assigned a serial number by data managers, to ensure anonymity.

According to our national guidelines, retrospective studies using anonymized data are exempted from ethical committee approval. Moreover, all the women enrolled signed an informed consent allowing data collection and analysis.

The CRB protocol was the same in each center, in agreement with Italian National Guidelines and subsequent hospitals’ protocols [12]. The CRB was placed transcervically using a speculum; after passing the internal orifice of the cervical canal, both balloons were filled with 60 mL sterile saline. Unless spontaneous expulsion occurred, the CRB was left in place for 24 h; upon its removal, the induction was continued by amniotomy whenever possible. In the case of absent or scarce uterine activity, intravenous low-dose oxytocin infusion was used to continue the induction, starting with 2 mU/min (12 mL/h) and increasing by 2 mU/min steps every 30 min, until the target of 5 contractions in 10 min was achieved.

Oxytocin was also cautiously administered with the purpose of augmenting uterine activity in the case of prolonged or arrested first and second stages of labor. According to the American College of Obstetricians and Gynecologists (ACOG) and the Society for Maternal-Fetal Medicine (SMFM) [13], first-stage labor progression was considered abnormal when the cervical dilatation, after reaching 6 cm, increased less than 0.5–0.7 cm/h for nulliparous and 0.5–1.3 cm/h for multiparous patients. The first stage of labor was defined as arrested when, together with cervical dilatation ≥6 cm, with ruptured membranes, there was no dilatation progress despite 4 h of adequate uterine activity or at least 6 h of oxytocin infusion. The second stage of labor was defined as arrested after at least 2 h of active pushing in multiparous and 3 h in nulliparous patients; one additional hour was allowed in case of epidural analgesia.

### 2.1. Outcomes

Maternal variables that were considered were the following: age, ethnicity, parity, mode of conception (natural or with assisted reproduction technology, ART), pre-pregnancy and term body mass index (BMI), indication for previous CS, indication for labor induction, gestational age at CRB insertion, amniorexis after CRB removal, oxytocin induction, oxytocin augmentation, epidural analgesia, mode of delivery, indication for CS or operative vaginal delivery, and presence of maternal complications. We also considered a composite adverse maternal outcome (CAMO), defined as the presence of one or more of the following major complications: post-partum hemorrhage (blood loss > 1000 mL), uterine rupture, need for laparotomy, need for hysterectomy, post-partum infection, and need for blood transfusion.

Fetal variables considered were the following: birthweight, incidence of pathological fetal heart rate (FHR) tracing during labor according to FIGO classification [14], Apgar score at 1 and 5 min, umbilical artery pH, and admission to neonatal intensive care unit (NICU). We also considered a composite adverse fetal outcome (CAFO), including major fetal complications.

The primary outcome of the study was the VBAC rate; secondary outcomes were the rates of pathological FHR tracing, Apgar score < 7 at 5 min, arterial pH < 7.1, NICU admission.

### 2.2. Statistical Analysis

The statistical package SPSS Statistics 21.0 (IBM SPSS Statistics, New York, NY, United States) was used for all the analyses. A Kolmogorov–Smirnov test was used to check the normal distribution of data; as most of them had a skewed distribution, they were expressed as median plus interquartile range (IQR) (continuous variables), or percentage (categorical variables). The chi square (χ^2^) test, Fisher exact test, unpaired Student’s *t*-test, or Mann–Whitney U test were used to compare differences among subgroups. The Pearson’s correlation analysis was used to assess the relationship among variables. A two-sided *p* value < 0.05 was considered significant. Variables found to be significant in univariate analysis were included in a stepwise multivariate logistic regression model. The associations between variables and successful TOLAC were presented as adjusted odds ratio (aORs) with corresponding 95% confidence intervals (95% CI).

## 3. Results

Overall, the database included 3574 women with previous CS. Among them, 765 were excluded because of more than one previous CS, 956 because of the spontaneous onset of labor, 752 due to having a Bishop score > 6 at hospital admission, 149 for the presence of contraindications to spontaneous labor, 23 for the presence of fetal malformation, and 10 for intrauterine fetal death. Moreover, 654 of the remaining women did not accept to undergo labor induction. Finally, 265 women met the inclusion criteria and were included in the study (Figure 1).

The main clinical characteristics of the women included in the study appear in Table 1. The indications for induction of labor were: post-term pregnancy (n = 94, 35.5%), diabetes (n = 47, 17.7%), hypertensive disorders (n = 39, 14.7%), intrauterine growth restriction and/or oligohydramnios (n = 20, 7.5%), intrahepatic cholestasis (n = 23, 8.6%), and other (n = 42, 15.8%).

Successful induction ending with vaginal birth (VBAC) occurred in 152 women (57.3%); another 14 women (9.2%) had vaginal delivery by obstetrical vacuum extractor. VBAC rate was significantly higher in women who already had a vaginal delivery (80.8% vs. 52.3% *p* < 0.001), and when intrapartum epidural analgesia was used (58.6% vs. 34.5% *p* < 0.001) (Table 2). On the contrary, pre-pregnancy overweight/obesity (BMI ≥ 30) and advanced age (≥40 years) were significantly associated with the failure of TOLAC and the need to repeat a CS (28.3% vs. 11.8% *p* = 0.001 and 15.9% vs. 7.2%, *p* = 0.025, respectively) (Table 2). The likelihood of VBAC was not influenced by the gestational age at induction, the mode of conception, ethnicity, the indication of previous CS, and the indication of labor induction (Table 2).

The median time from the application of the balloon catheter to the onset of labor was significantly shorter in the VBAC group than in the CS group (16, IQR 12–22 h vs. 19, IQR 16–25; *p* = 0.018), whereas the median time from the application of the CRB to delivery was comparable in the two groups (21, IQR 15–27 vs. 22, IQR 17–31; *p* = 0.207) (Table 2). The use of oxytocin for induction of labor did not increase the rate of delivery (*p* = 0.949), while if administered for augmentation, during the second stage of labor, it was associated with a higher rate of successful VBAC (32.2% vs. 21.2%, *p* = 0.047). Intrapartum analgesia was associated with an increased success of vaginal birth (69.5% vs. 45.9% *p* = 0.000).

The following variables remained statistically significant in the multivariate logistic regression model: history of vaginal delivery (aOR 1.94, 95% CI 1.23–3.64), maternal age ≥ 40 years (aOR 1.22, 95% CI 1.04–1.67), pre-pregnancy BMI (aOR 1.81, 95% CI 1.33–3.36).

The incidence of composite adverse maternal outcome (CAMO) was comparable in the VBAC group and in the CS group (12.5% vs. 8%; *p* = 0.235), whereas the incidence of composite adverse fetal outcome (CAFO) was significantly lower after vaginal delivery (3.3% vs. 12.4%; *p* = 0.005) (Table 2). Interestingly, the incidence of CAMO was 4.8% among women induced with a CRB alone vs. 17.6% among those induced with a CRB plus oxytocin (*p* = 0.001), whereas CAFO had comparable incidence (4.8% vs. 10.1% *p* = 0.097) (Table 3). Table 3 shows the incidence of each single adverse maternal outcome and of each adverse fetal outcome; of note, uterine rupture occurred in one case (0.4%) in the CRB plus oxytocin group. In this unique case of uterine rupture, oxytocin was administered with the purpose of augmentation.

## 4. Discussion

In this study, TOLAC using a CRB in patients with a previous CS and unfavorable cervix resulted in a VBAC rate of 57.3%, which is consistent with the previously published average success rate [15]. We observed a positive relationship between having already delivered vaginally and VBAC: more than 80% of women with a prior spontaneous delivery could give birth vaginally after CS; this observation is consistent with recently published data showing that a prior vaginal delivery and/or a prior successful VBAC are the strongest predictors of a successful TOLAC [7]. We also found a significant positive effect of epidural analgesia on VBAC rate, as reported in recent guidelines [16]; indeed, epidural analgesia is recommendable in TOLAC, and adequate pain control may encourage more women to choose induction of labor instead of elective CS [10,11].

On the other side, women above 40 years and with a pre-pregnancy BMI > 30 were found to be at higher risk of TOLAC failure, with the need to undergo another CS. Indeed, a published prediction model for TOLAC success showed that the risk of repeating CS increases linearly with the woman’s age and with BMI [5]. Furthermore, a previous study showed that women with normal BMI during the first pregnancy, who became overweight or obese before the second pregnancy, had a significantly lower likelihood of successful VBAC compared with women with unchanged normal BMI in the second pregnancy [17].

Although some studies claimed that anticipating the induction of labor at 39 weeks in low-risk nulliparous women could significantly lower the frequency of CS [16], we did not observe any advantage in performing TOLAC at 39 weeks, as the rate of VBAC was not influenced by the gestational age at induction. According to our data, there is no need to anticipate the time of delivery to reduce the risk of repeated CS.

In the present study, induction with CRB was followed by oxytocin infusion in about 45% of cases; accordingly, previous studies in women with an unscarred uterus showed that CRB application is associated with a significantly higher need for oxytocin compared to prostaglandins [18], and a CRB alone is able to induce the onset of labor in no more than 30% of women [19]. In a recent systematical review on the application of CRBs at term after previous CS, the average proportion of women requiring oxytocin was 68.4%, ranging from 20.5% to 91.5% [15]. Oxytocin use for labor induction in women with an unfavorable cervix was reported to be associated with an increased risk of CS [20,21]; we found no significant difference in VBAC rate between women who received oxytocin and women who did not. Differently, the use of oxytocin, not for inducing labor, but just for augmenting uterine activity after the onset of labor, led to a higher chance of VBAC.

Concerning adverse maternal outcomes, we did not find any difference in the CAMA incidence between women delivering vaginally and those undergoing a repeated CS; we observed, however, that patients experiencing both CRB application and oxytocin infusion showed a higher CAMA incidence; of note, the only case of uterine rupture occurred in a woman treated with both a CRB and oxytocin. As previously reported, oxytocin infusion itself is associated with a significantly greater risk of uterine rupture in patients with a previous CS [8]: a recent study comparing CRB vs. oxytocin found that CAMA incidence was higher in women receiving oxytocin, with an odds ratio of 1.43 [22,23].

The CAFO incidence rate appeared to be higher in patients needing repeated CS and/or needing oxytocin to complete TOLAC; the higher CAFO incidence rate in the case of oxytocin use is likely related to uterine hyperstimulation with negative effects on FHR [24,25]. A careful approach to oxytocin use is mandatory to reduce neonatal complications; a recent metanalysis found that the administration of oxytocin after CRB use did not reduce CS rate, and should be carefully evaluated [26].

### 4.1. Strengths and Limitations

The strength of this study can be found in the strict criteria of inclusion, the multicenter design, and the complete follow-up of all studied women. Its limitations are the retrospective nature and the relatively small sample size, that was in part overcome by analyzing composite outcomes. A further limitation is the lack of comparison with a similar group of women induced with other methods, secondary to the concern of the use of pharmacological methods in pregnancies when attempting TOLAC. Indeed, our national guidelines, in agreement with other international societies [10], are hostile to the use of prostaglandins or other drugs in case of labor induction of women with a previous CS. As a consequence, induction with a CRB is the only option for labor induction in women with an unfavorable cervix.

### 4.2. Implications for Clinical Practice

Our findings may be useful for clinicians taking care of women with one previous CS and an unfavorable cervix, as they demonstrate the efficacy and safety of labor induction with a CRB. According to our data, TOLAC should be considered even in the case of an unfavorable Bishop score if adequate fetal and maternal monitoring can be provided, and emergency CS can be rapidly performed as needed. Moreover, this study provides a warning on the use of oxytocin in the case of augmentation, suggesting a judicious administration. Overall, performing TOLAC with a CRB in patients with one previous CS and an unfavorable cervix is cheap and is associated with an acceptable success rate and safety profile.

## 5. Conclusions

In women with a previous CS and an unfavorable cervix, induction of labor with a CRB has an encouraging success rate in terms of VBAC. The accurate selection of candidates is the key point, together with the use of epidural analgesia. Maternal age over 40, pre-pregnancy BMI over 30, and having no previous vaginal birth are major risk factors for TOLAC failure. A combined use of a CRB and oxytocin for induction of labor increases the rate of adverse maternal outcome without improving the VBAC likelihood.

## Figures and Tables

**Figure 1 healthcare-11-00543-f001:**
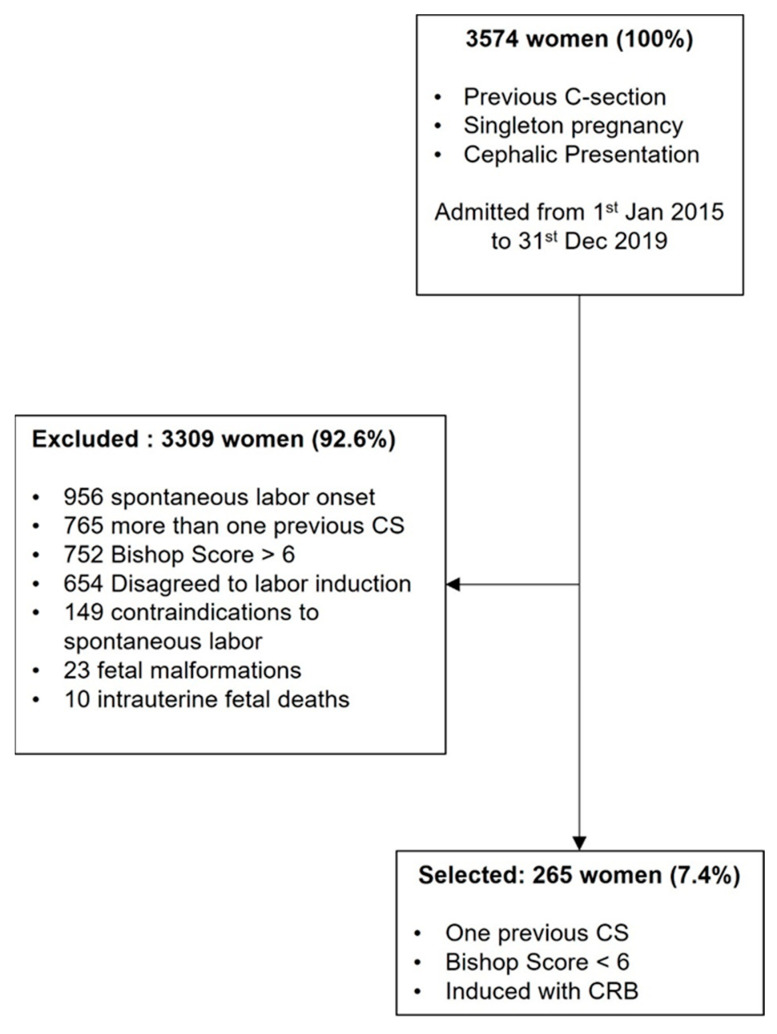
STROBE flow chart. CS: cesarean section.

**Table 1 healthcare-11-00543-t001:** General clinical characteristics of the women included in the study.

Age (Years)	34 (29–37)
Ethnicity	
Caucasian	208 (78.6%)
Black	24 (9%)
Asian	21 (7.9%)
Hispanic	12 (4.5%)
Previous vaginal birth	
Yes	47 (17.7%)
No	218 (82.3%)
Gestational Age at Induction (weeks)	40 (38–41)
BMI	
Pre-pregnancy	24.2 (21.6–27.8)
Term pregnancy	28.9 (26.2–32.4)
Mode of conception	
Spontaneous	261 (98.5%)
ART	4 (1.5%)

Data are expressed as median and interquartile range (IQR) or as number and percentage. BMI: body mass index; ART: assisted reproduction technique.

**Table 2 healthcare-11-00543-t002:** Characteristics of women undergoing labor induction and related labor outcomes: comparison between 152 women delivering by vaginal birth (VBAC) vs. 113 requiring CS.

	VBAC n = 152	CS n = 113	*p*
Maternal age (years)	34 (29–36)	35 (30–38)	0.479
Maternal age ≥40	7.2%	15.9%	0.025
Gestational age at induction (weeks)	40 (38–41)	40 (39–41)	0.101
Gestational age ≥39 weeks	55.9%	61.1%	0.402
Mode of conception			0.764
Spontaneous	150 (98.7%)	111 (98.2%)	
ART	2 (1.3%)	2 (1.8%)	
Previous vaginal birth	25%	8%	<0.001
Ethnicity			0.395
Caucasian	121 (79.6%)	87 (77%)	
Black	13 (8.6%)	11 (9.7%)	
Asian	12 (7.9%)	9 (8%)	
Hispanic	6 (3.9%)	6 (5.3%)	
Pre-pregnancy BMI	23.3 (21–26)	25.6 (22.9–30.6)	<0.001
≥30	11.8%	28.3%	0.001
Term-pregnancy BMI	27.7 (25.5–30.5)	30.6 (27.5–34.2)	0.054
Indication of labor induction			0.386
Post-term pregnancy	49 (32.2%)	45 (39.8%)	
Diabetes	30 (19.7%)	17 (15%)	
Hypertensive disorders	21 (13.8%)	18 (15.9%)	
IUGR/oligohydramnios	10 (6.6%)	10 (8.9%)	
Intrahepatic cholestasis	13 (8.6%)	10 (8.9%)	
Others	29 (19.1%)	13 (11.5%)	
Indication for previous CS			0.210
Breech presentation	30 (19.7%)	18 (37.5%)	
Dystocia	18 (11.8%)	21 (53.8%)	
Pathological FHR tracing	25 (16.4%)	21 (45.7%)	
Failed labor induction	10 (6.6%)	15 (60%)	
Maternal will	1 (0.7%)	0 (0%)	
Others	47 (30.9%)	28 (37.3%)	
Unknown	21 (13.8%)	10 (32.3%)	
Duration of CRB application (h)	12 (9–15)	13 (12–16)	0.285
Time of CRB application—labor onset (h)	16 (12–22)	19 (16–25)	0.018
Time of CRB application—delivery (h)	21 (15–27)	22 (17–31)	0.207
CRB plus oxytocin induction	44.7%	45.1%	0.949
Oxytocin augmentation	32.2%	21.2%	0.047
Use of intrapartum analgesia	58.6%	34.5%	<0.001
Incidence of CAMO	12.5%	8%	0.235
Incidence of CAFO	3.3%	12.4%	0.005

Data are expressed as median and interquartile range (IQR) or as number and percentage. ART: assisted reproduction technique; BMI: body mass index; IUGR: intrauterine growth restriction; CAMO: composite adverse maternal outcome; CAFO: composite adverse fetal outcome.

**Table 3 healthcare-11-00543-t003:** Incidence of composite adverse maternal (CAMO) and fetal outcomes (CAFO) in women induced with CRB alone (n = 146) or with CRB plus oxytocin (n = 119).

	CRB n = 146	CRB Plus Oxytocin n = 119	*p*
Incidence of CAMO	7 (4.8%)	21 (17.6%)	0.001
Uterine rupture	0	1 (0.8%)	
Post-partum hemorrhage	4 (2.7%)	6 (5%)	
Endometritis	0	1 (0.8%)	
Wound re-opening	1 (0.68%)	1 (0.8%)	
Thromboembolism	2 (1.36%)	0	
Surgical lesions	0	1 (0.8%)	
Others (laparotomy, hysterectomy)	0	3 (2.5%)	
Blood transfusion	0	5 (4.2%)	
Intensive care unit admission	0	3 (2.5%)	
Incidence of CAFO	7 (4.8%)	12 (10.1%)	0.097
Neonatal intensive care unit admission	1 (0.68%)	4 (3.3%)	
Apgar score (5′) < 7	2 (1.36%)	3 (2.5%)	
Umbilical pH < 7	1 (0.68%)	2 (1.68%)	
Pathological FHR tracing	3 (2%)	3 (2.5%)	

## Data Availability

The data presented in this study are available on request from the corresponding author. The data are not publicly available due to privacy policies.

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
