# Peer review of "Induction of Labor in Women with Previous Cesarean Section and Unfavorable Cervix: A Retrospective Cohort Study"

_healthcare, 2023, doi:10.3390/healthcare11040543_

Round 1

Reviewer 1 Report

The authors performed a retrospective cohort study to investigate the outcomes of induction of labour by cervical ripening balloon in 265 pregnant women with one previous Caesarean section and an unfavourable cervix. They concluded that this induction method is safe and effective. This manuscript is well written, including several important variables. In my opinions, a revision is required to clarify some points before acceptance for publication.  

Introduction

Lines 40-41. Would the authors revise this sentence as the meaning is not very clear?

Line 55. Would the authors revise the word ‘large’? The sample size was 265, not large.

Methods

Line 61. Would the authors use the word ‘decline’ instead of ‘refused’?

Lines 71-79. Would the authors provide references to this CRB protocol?

Line 78. Is five uterine contractions/ min too frequent?

Lines 80-81. The use of oxytocin in the second stage of labour should be cautious, especially in women with a previous C section.

 Results

Lines 117-124. Would the authors avoid duplication of the text with Figure 1?

Lines 135-139. Would the authors perform logistic regression to control the confounding factors which affect successful VBAC and present the adjusted Odd ratios?  

Lines 163-164. Would the authors add whether the oxytocin was used for induction or augmentation in the case with uterine rupture?

Table 3. Would the authors revise the words ‘Wound opening’ and ‘surgical lesions’? Do the authors mean ‘gap wound’ and ‘injuries’?

Discussion

Lines 200-201. Would the authors explain why the use of oxytocin for augmentation of labor was associated with a higher chance of VBAC? This observation is not consistent with the use in women without uterine scar?

Lines 218-220. Would the authors add ‘no control group (other methods of induction or ERCS)’  in the limitations?

Lines 222-228. Would the authors add ‘cautious use of oxytocin’ and ‘long induction -delivery interval’?

References

Lines 266-268, Lines 307-309. Would the authors revise the reference list and the numbers in the text?

Author Response

Answer to Reviewer 1

First, we want to thank Reviewer 1 for time she/he dedicated to revise our manuscript and for issues she/he has brought to light.

The authors performed a retrospective cohort study to investigate the outcomes of induction of labour by cervical ripening balloon in 265 pregnant women with one previous Caesarean section and an unfavourable cervix. They concluded that this induction method is safe and effective. This manuscript is well written, including several important variables. In my opinions, a revision is required to clarify some points before acceptance for publication.  

Introduction

Lines 40-41. Would the authors revise this sentence as the meaning is not very clear?

We clarified the concept in a more complete sentence.

Lines 40-44. “…on the other side, however, women who experienced vaginal birth after previous cesarean section (VBAC) are demonstrated less likely to face birth-related morbidity such as blood transfu-sion, uterine rupture, unplanned hysterectomy, and admission to the Intensive Care Unit than women who have ERCS.”

Line 55. Would the authors revise the word ‘large’? The sample size was 265, not large.

Accepted. We changed large with fair-sized.

Lines 62-64. “Therefore, the objective of the present study was to retrospectively evaluate the efficacy and safety of CRB for labor induction in a fair-sized cohort of women with a previous CS and unfavorable cervix.”

Methods

Line 61. Would the authors use the word ‘decline’ instead of ‘refused’?

Accepted.

Line 69. “decline to labor induction”

Lines 71-79. Would the authors provide references to this CRB protocol?

We added reference (12)

  1. Fondazione Confalonieri Ragonese on behalf of SIGO, AOGOI, AGUI, Induzione al travaglio di parto, June 2016, revised February 2022.

Line 78. Is five uterine contractions/ min too frequent?

It is not too frequent but the target we want to achieve in case of labor induction/augmentation, in order to avoid fetal distress.

Lines 88-89 “…until the target of 5 contractions in 10 minutes was achieved.”

Lines 80-81. The use of oxytocin in the second stage of labour should be cautious, especially in women with a previous C section.

We are aware of this; all of the drugs used for labor induction are at risk for uterine rupture, however oxytocin is the one considered to have a slightly lower risk profile of uterine rupture during TOLAC and it is the unique trump card in case of absence of uterine contractile activity. In any case, we agreed to stress this point and we provided to complete the sentence including this consideration.

Lines 90-91: Oxytocin was also cautiously administered with the purpose of augmenting uterine activity in case of prolonged or arrested first and second stage of labor. 

 Results

Lines 117-124. Would the authors avoid duplication of the text with Figure 1?

Thanks for the suggestion; however, we prefer to have both.

Lines 135-139. Would the authors perform logistic regression to control the confounding factors which affect successful VBAC and present the adjusted Odd ratios?  

Thanks for the suggestion. We performed multivariate logistic regression on significant variables.

Lines 139-142: “Variables found to be significant in univariate analysis were included in a stepwise multivariate logistic regression model. The associations between variables and successful TOLAC were presented as adjusted Odds ratio (aORs) with corresponding 95% confidence intervals (95%CI).”

Lines 180-183: “The following variables remained statistically significant in the multivariate logistic regression model; history of vaginal delivery (aOR 1.94 95%CI 1,23-3,64), maternal age > 40 years (aOR 1.22 1.04-1.67), pre-pregnancy BMI (aOR 1.81 95%CI 1.33-3.36)”.

Lines 163-164. Would the authors add whether the oxytocin was used for induction or augmentation in the case with uterine rupture?

In the unique case of uterine rupture we faced, the oxytocin was used for augmentation. We provided to add this information to the manuscript.

Lines 174-176 “of note, uterine rupture occurred in one single case (0.4%) in the CRB plus oxytocin group: in this unique case uterine rupture, oxytocin was administered with the purpose of augmentation.”

Table 3. Would the authors revise the words ‘Wound opening’ and ‘surgical lesions’? Do the authors mean ‘gap wound’ and ‘injuries’?

With “wound opening” we intend the need for re-operation after C-section, with the consequent re-opening of the surgical wound, for many reasons including dehiscence of the surgical suture or wound hematoma. With surgical lesions we intend ureter lesions, bladder lesions or gut lesions which are the most frequent site of lesion in case of repeated C-section. We modified “wound opening” with “wound re-opening” in order to be clearer.

Discussion

Lines 200-201. Would the authors explain why the use of oxytocin for augmentation of labor was associated with a higher chance of VBAC? This observation is not consistent with the use in women without uterine scar?

We explained this consideration in the Results section. Lines 168-170 “The use of oxytocin after CRB for labor induction, performed in 119 women, did not increase the VBAC rate (p=0.949); differently, when oxytocin was given to 146 women to augment uterine activity, VBAC was significantly facilitated (p=0.047)”

Lines 218-220. Would the authors add ‘no control group (other methods of induction or ERCS)’  in the limitations?

Thanks for your comment. According to our national guidelines and ACOG recommendations (ACOG Practice Bulletin No. 205: Vaginal Birth After Cesarean Delivery. Obstet Gynecol. 2019 Feb;133(2):e110-e127. doi: 10.1097/AOG.0000000000003078) due to the higher risk of uterine rupture, the use of pharmacological agents, such as misoprostol or prostaglandin, is not recommended. As a consequence, a comparison is not possible. This limitation was now acknowledged in the discussion.

Lines 266-268: “ A further limitation is the lack of comparison with a similar group of women induced with other methods secondary to the concern of the use of pharmacological methods in pregnancies attempting TOLAC.”

Lines 222-228. Would the authors add ‘cautious use of oxytocin’ and ‘long induction -delivery interval’?

Accepted.

Lines 252-253 “Moreover, this study provide a warning on the oxytocin use in case of augmentation, suggesting a judicious administration.”

References

Lines 266-268, Lines 307-309. Would the authors revise the reference list and the numbers in the text?

Revised.

Reviewer 2 Report

In this article, the authors present a retrospective study analyzing the efficacy and safety of the use of cervical ripening balloon in the induction of labor in pregnant women with a previous cesarean section. The article is interesting given the scarce evidence on the subject, and supports a clinical practice endorsed by the main scientific societies.  The results of the study also support this practice, showing composite adverse fetal outcomes significantly lower after vaginal delivery.

It would have been interesting if the authors had analyzed the cervical conditions by exploration (BISHOP) or ultrasonography, in order to establish by multivariate analysis the factors that lead to a higher success rate. However, I understand that this is not the aim of this study, but I recommend that they explore this way.

It is also a well-targeted and well-structured article. However, before publication, I believe that the following minor considerations should be considered and corrected in relation to the manuscript:

1. Introduction (lines 34-35): "The best delivery mode for woman with a previous caesarean section (CS) is still matter of discussion, and clinical practice varies worldwide". Although clinical practice may vary between different countries and centers, undoubtedly the mode of delivery of choice after a previous cesarean section is the vaginal route, and I believe that this should be made clear in this article from the outset. Obviously, this does not conflict with the fact that the cesarean rate in this group of patients is high, or that the likelihood of vaginal delivery is lower than in patients without prior cesarean.

2. Materials and methods (line 57): it is striking that among the exclusion criteria, gestational age was not taken into account. I do not criticize it, but I would like to know why the authors have considered it this way.

3. Materials and methods (line 57): I am also struck by the fact that they do not refer to Robson's classification, since it seems that they stick to Group 5.

4. Materials and methods (lines 67-68): "According to national guidelines, the study did not need any Ethical Committee approval". I am not familiar with the Italian regulations in this respect, so I would ask the authors to inform me about this, as this aspect is critical before publication. If not, being a retrospective study, it would require either the approval of the ethical committee or at least to be informed by the committee of the absence of the need for its evaluation.

5. Materials and methods (lines 107-108): it would be advisable for the outcomes to appear in a specific section, outside Statistical Analysis.

In conclusion, the authors present a retrospective analysis of a cohort of patients with previous cesarean section who underwent induction of labor by mechanical methods. The results are interesting, and support the use of this technique, so I believe it can be considered for publication after a minor revision.

Author Response

Answer to Reviewer 2

First, we want to thank Reviewer 1 for time she/he dedicated to revise our manuscript and for issues she/he has brought to light.

In this article, the authors present a retrospective study analyzing the efficacy and safety of the use of cervical ripening balloon in the induction of labor in pregnant women with a previous cesarean section. The article is interesting given the scarce evidence on the subject, and supports a clinical practice endorsed by the main scientific societies.  The results of the study also support this practice, showing composite adverse fetal outcomes significantly lower after vaginal delivery.

It would have been interesting if the authors had analyzed the cervical conditions by exploration (BISHOP) or ultrasonography, in order to establish by multivariate analysis the factors that lead to a higher success rate. However, I understand that this is not the aim of this study, but I recommend that they explore this way.

It is also a well-targeted and well-structured article. However, before publication, I believe that the following minor considerations should be considered and corrected in relation to the manuscript:

  1. Introduction (lines 34-35): "The best delivery mode for woman with a previous caesarean section (CS) is still matter of discussion, and clinical practice varies worldwide". Although clinical practice may vary between different countries and centers, undoubtedly the mode of delivery of choice after a previous cesarean section is the vaginal route, and I believe that this should be made clear in this article from the outset. Obviously, this does not conflict with the fact that the cesarean rate in this group of patients is high, or that the likelihood of vaginal delivery is lower than in patients without prior cesarean.

Thanks for the suggestion, however we prefer to guide the reader through data and evidence, showing what benefits and risks are related to labor induction and why is still controversial notwithstanding international guidelines suggestions. This logic is, according to us, useful in order to introduce the fact that the major problem we face with this group of women is the unfavorable Bishop score, that discourages the physician. 

  1. Materials and methods (line 57): it is striking that among the exclusion criteria, gestational age was not taken into account. I do not criticize it, but I would like to know why the authors have considered it this way.

Thanks for the issue. It’s an oversight, we fix it. We followed Guidelines indications.

Line 67: ” including women with singleton term pregnancy”

  1. Materials and methods (line 57): I am also struck by the fact that they do not refer to Robson's classification, since it seems that they stick to Group 5.

Indeed, we considered women in Robson’s Group 5, and we added this info.

Line 68: (Robson Classification - Group 5)

  1. Materials and methods (lines 67-68): "According to national guidelines, the study did not need any Ethical Committee approval". I am not familiar with the Italian regulations in this respect, so I would ask the authors to inform me about this, as this aspect is critical before publication. If not, being a retrospective study, it would require either the approval of the ethical committee or at least to be informed by the committee of the absence of the need for its evaluation.

As requested, we rephrased.

Lines 80-82: “According to our national guidelines, retrospective studies using anonymized data were exempted by ethical committee approval. Moreover, all the women enrolled signed an informed consent allowing data collection and analysis.”

  1. Materials and methods (lines 107-108): it would be advisable for the outcomes to appear in a specific section, outside Statistical Analysis.

Thanks for the suggestion. We add a new section: Outcomes.

In conclusion, the authors present a retrospective analysis of a cohort of patients with previous cesarean section who underwent induction of labor by mechanical methods. The results are interesting, and support the use of this technique, so I believe it can be considered for publication after a minor revision.

Thank you very much for your support and the interesting suggestions.

Reviewer 3 Report

Chiara Germano et al performed this retrospective cohort study on induction of labor in women with previous cesarean section (CS) and unfavorable cervix, and they found in women with a previous CS and unfavorable Bishop score, induction of labor with cervical ripening balloon (CRB) can be considered safe and effective.

Labor induction with mechanical devices as CRB, especially in case of unfavorable cervix, have been widely used in no CS history patients. But the efficacy and safety of CRB use after CS kept controversial. This study collected six tertiary hospitals data, which included 265 women used CRB and did some statistic in these 265 CRB used patients and try to give an answer of the CRB efficacy and safty.

1.     The study design has a problem. If we want to verify the CRB is useful, we should compare the same patients one group used CRB and another group used other labor induction such as pharmacological induction, then we check the VBAC rate of both group and make a conclusion of CRB is useful or not. But in the present study, all the 265 patients only used CRB, without comparison. From the present data, we can not make a conclusion CRB is more useful than other induction methods.

2.     Abbreviated words should be completely defined when first used, such us BS, CAMO and CAFO in the manuscript.

3.     In the abstract, 256 patients (line 22) included, but in methods and results, 265 patients are included.

4.     In the abstract, augmentation improved 22 vaginal delivery (32.2% vs 21.2%). But there is not data showed this in the results part.

5.     Many reference are strange, such as 5,7,22,23.

Author Response

Answer to Reviewer 3

First, We would like to thank Reviewer 1 for time he/she spent to review our manuscript. We are very grateful.

Chiara Germano et al performed this retrospective cohort study on induction of labor in women with previous cesarean section (CS) and unfavorable cervix, and they found in women with a previous CS and unfavorable Bishop score, induction of labor with cervical ripening balloon (CRB) can be considered safe and effective.

Labor induction with mechanical devices as CRB, especially in case of unfavorable cervix, have been widely used in no CS history patients. But the efficacy and safety of CRB use after CS kept controversial. This study collected six tertiary hospitals data, which included 265 women used CRB and did some statistic in these 265 CRB used patients and try to give an answer of the CRB efficacy and safty.

  1. The study design has a problem. If we want to verify the CRB is useful, we should compare the same patients one group used CRB and another group used other labor induction such as pharmacological induction, then we check the VBAC rate of both group and make a conclusion of CRB is useful or not. But in the present study, all the 265 patients only used CRB, without comparison. From the present data, we can not make a conclusion CRB is more useful than other induction methods.

Thanks for your comment. According to our national guidelines and ACOG recommendations (ACOG Practice Bulletin No. 205: Vaginal Birth After Cesarean Delivery. Obstet Gynecol. 2019 Feb;133(2):e110-e127. doi: 10.1097/AOG.0000000000003078) due to the higher risk of uterine rupture, the use of pharmacological agents, such as misoprostol or prostaglandin, is not recommended. As a consequence, a comparison is not possible. This limitation was now acknowledged in the discussion.

Lines 266-268: “ A further limitation is the lack of comparison with a similar group of women induced with other methods secondary to the concern of the use of pharmacological methods in pregnancies attempting TOLAC.”

  1. Abbreviated words should be completely defined when first used, such us BS, CAMO and CAFO in the manuscript.

Thanks for suggestion, however we already provided to define them.

Lines 170-173: “The incidence of Composite Adverse Maternal Outcome (CAMO) was comparable in the VBAC group and in the CS group (12,5% vs. 8%; p=0.235), whereas the incidence of Composite Adverse Fetal Outcome (CAFO) was significantly lower after vaginal delivery (3.3% vs. 12,4%; p=0.005) (Table 2).”

  1. In the abstract, 256 patients (line 22) included, but in methods and results, 265 patients are included.

Thanks. It is an oversight. Patients enrolled were 265, we fixed it.

  1. In the abstract, augmentation improved 22 vaginal delivery (32.2% vs 21.2%). But there is not data showed this in the results part.

Thank you. Data are shown in table 2. However, we added a sentence about this finding.

Lines 169-172: “The use of oxytocin for induction of labor didn’t increase the rate of delivery (p=0.949), while if ad-ministered for augmentation, during the second stage of labor, it was associated to a high-er rate of successful VBAC (32.2% vs 21.2%, p=0.047). Intrapartum analgesia was associated to an increase success of vaginal birth rate (69.5% vs 45.9% p=0.000).”

  1. Many reference are strange, such as 5,7,22,23.

Thank you. We fixed them.

Round 2

Reviewer 3 Report

Thank you for the reply.

Although the authors said the high risk of pharmacological induction, I recommended that compare the CRB group with other group your institution used before, not only show the CRB used group results.

Author Response

Thank you for the reply.

Although the authors said the high risk of pharmacological induction, I recommended that compare the CRB group with other group your institution used before, not only show the CRB used group results.

Thanks again for your consideration and suggestion about our manuscript.

As already reported national guidelines, in agreement with other international guidelines (e.g. ACOG), contraindicated the use of prostaglandin or other pharmacological agents in women with a previous CS. As a consequence, CRB is the only method allowed to induce labor in women with unfavorable cervix. This concept has been further explained in the discussion.

Lines 267-273 “A further limitation is the lack of comparison with a similar group of women induced with other methods, secondary to the concern of the use of pharmacological methods in pregnancies attempting TOLAC. Indeed, our national guidelines, in agreement with other international societies (10), are hostile to the use of prostaglandins or other drugs in case of labor induction of women with a previous CS. As a consequence, induction with CRB is the only option for labor induction in women with unfavorable cervix.”